# Design of Wearable Finger Sensors for Rehabilitation Applications

**DOI:** 10.3390/mi14040710

**Published:** 2023-03-23

**Authors:** Beyza Bozali, Sepideh Ghodrat, Kaspar M. B. Jansen

**Affiliations:** Faculty of Industrial Design Engineering, Delft University of Technology, 2628 CE Delft, The Netherlands

**Keywords:** knitted strain sensor, rehabilitation applications, smart textiles, wearable textiles

## Abstract

As an emerging technology, smart textiles have attracted attention for rehabilitation purposes or to monitor heart rate, blood pressure, breathing rate, body posture, as well as limb movements. Traditional rigid sensors do not always provide the desired level of comfort, flexibility, and adaptability. To improve this, recent research focuses on the development of textile-based sensors. In this study, knitted strain sensors that are linear up to 40% strain with a sensitivity of 1.19 and a low hysteresis characteristic were integrated into different versions of wearable finger sensors for rehabilitation purposes. The results showed that the different finger sensor versions have accurate responses to different angles of the index finger at relaxation, 45° and 90°. Additionally, the effect of spacer layer thickness between the finger and sensor was investigated.

## 1. Introduction

Flexible and wearable sensors have attracted attention in recent years for a variety of applications, including human–device interfaces and health monitoring, such as respiration rate, heart rate, and body position [1,2,3]. Traditional sensors are often integrated into garments as an external element or attached to the surface, and these create discomfort for the user due to the bulky and rigid nature of electronic devices, such as inertial measurement units (IMUs) for health monitoring purposes [4]. Traditional sensors based on rigid electronics are usually bulky and uncomfortable during movement and not always suited for long-term use and need to be adapted for applications in at-home situations. Moreover, rigid to soft transitions are vulnerable to failures due to fatigue during wearing and washing [5]. For future generations, it is desirable if wearable sensors would be flexible and stretchable, such that they conform to the body contours, as well as being breathable and washable. As an example, textile-based sensors offer a new possibility for fully integrated monitoring of body-related parameters, such as heart rate or local strains. They are easy to wear, flexible, and washable, and can be used for numerous health monitoring applications, making them a good alternative to traditional bulky sensors [6].

Knitted strain sensors are obtained by embedding conductive yarns in the knit structures and can be manufactured with commercially available conductive yarns and existing standard textile manufacturing equipment. They are completely integrated into the textile and can measure stains up to 40%. That makes them good candidates for unobtrusive monitoring of body posture and movement of hands, arms, legs, and fingers as they provide a good integration into garments. Additionally, they can apply this to measure limb movement during a therapy session and give feedback to the patient and therapists [7]. In literature, textile strain sensors have been produced either (i) by inserting stretchable sensor yarns produced at the laboratory conditions; (ii) by coating an existing knitted structure with conductive materials, such as graphene [8,9,10], PEDOT:PSS [9,11], or nanosilver [12]; and (iii) by introducing commercially conductive yarns during the knitting process. Although strain sensors with conductive materials have good stability, high sensitivity, and fast response, they lack flexibility, which hinders their stretchability under conditions of large deformation. This is because the conductive component may have structural damage as a result of high friction. Additionally, their mechanical incompatibility with textiles subsequently causes the electrical properties of the sensors to deteriorate and difficult to scale up [13,14]. As an alternative, these challenges can be realized by benefiting from using conductive yarns in knitted strain sensors.

Knitted strain sensors are 3D structures where plains of conductive yarns result in measurable resistance changes. The sensor mechanism in knitted structures is related to contact resistance between the conductive yarns under deformation. While the interlocked loops are separated from each other, contact resistance between successive knitted loops of conducting yarn changes which result in a decrease or increase in resistance [15].

Knitted strain sensors are quite suitable for rehabilitation applications and many studies have been reported so far. Ryu et al. investigated the performance of the knitted strain sensor in a glove by using silver-plated yarns to distinguish the finger movements and the electrical responses of the compressive strain demonstrated strong stability and linearity through various finger rolling angles [16]. Different hand motions were also detected by the knitted sensor developed by Han et al., but the electrical resistance change was not high enough to monitor the overall movements of the fingers [1]. Another study also studied the performance of the sensor in gloves for recognizing gestures and concluded that elasticity of the fabric affects the effective detection range of knitted sensors, the size of the sensor has a significant influence on sensitivity, and appropriate glove size helps to avoid non-linear sensing phenomenon [17]. Lee et al. also concluded that the developed glove might be useful to amputees as a tool that allows them to rehabilitate or regulate the myoelectric prosthesis by putting the sensing elements into the glove and producing the whole garment knitting technique for ease of commercialization [18]. Isaia et al. evaluated the performance of strain sensors knitted with various conductive yarns in terms of sensing properties, hysteresis and comfort for joint motion tracking applications during repetitive flexion-extension cycles [19,20].

Although it is feasible to produce knitted strain sensors with different designs using conductive yarns, performance is not always optimal resulting in a limited working range, non-linearity, and hysteresis. That results in poor sensing performance and they are currently not commonly integrated into practical applications. To be able to apply the knitted strain sensors in practical applications, first of all, the sensor itself must have a suitable working range, good linearity, and low resistance [20]. In addition, it should be realized that the textile structures in which the sensor performance is integrated are also affected by thickness, elasticity, and slipping of the fabric.

In this study, we focus on the application of a knitted strain sensor into a finger sensor for movement detection. We previously developed a knitted strain sensor with a working range of 40% and sensitivity of 1.19 ± 0.03 GF and low hysteresis of 0.03 (smaller than 1%) [21]. Additionally, the knitted strain sensor is integrated into different versions of wearable finger sensors. Several finger sensor designs will be considered and their sensor performance will be evaluated on a finger movement sensor and its use.

## 2. Materials and Methods

In a previous study [21], a textile-based strain sensor and its electromechanical performance were analyzed, and in the current study, this newly developed strain sensor was integrated into a finger sensor for the detection of movement. The strain sensor was knitted on a Stoll CMS 530 machine with a 1 × 1 Rib design by using conductive yarn from Shieldex with a yarn count of dtex 235 and initial resistance of ≤600 Ω/m and elastic yarn from Yeoman of Nm 15 as illustrated in Figure 1. The plating technique was used to position the conductive yarn inside and the elastic yarn outside in the sensing region (See Figure 1b,c). The conductive yarn is knitted together with a non-conductive elastic yarn using the two eyes of the plating carrier. Thus, the position of the conductive yarn relative to the elastic support yarn is well controlled, resulting in a rib structure in which the conductive yarn is positioned always inside. During stretching, conductive loops inside slide over each other resulting in a measurable resistance change.

The knitted strain sensor is integrated into different versions of wearable finger sensors, as shown in Figure 2. Version 1 is the knitted sensor strip attached to a non-woven tube fabric as shown in Figure 2a, whereas Version 2 is the knitted sensor strip attached between the fingertip and wrist. We expect that because of the small pre-stress in the Version 2 sensor, the slack is reduced during operation. In the version 1 sensor, we add a spacer layer of different thicknesses and we expect this to increase the sensor stretching and thus a measured signal. The advantages and disadvantages of the different versions are listed in Table 1. The length and width of the knitted strain sensor were set to 8 cm and 2 cm, respectively. Apart from the knitted sensor, the design includes a Arduino Nano BLE and a power supply with an integrated USB connector.

Before the application of the strain sensor into a finger wearable demonstrator, the characteristics of the knitted strain sensor strips were tested in the course-wise direction by performing four test cycles at 30 mm/min, using a custom-made tensile tester and the resistance response during tensile extension–relaxation tests was assessed. The gauge length was set as 50 mm in the customized tensile tester. The tests were conducted up to 40% strain considering the finger deformation is less than 20 mm extension [22]. To evaluate the electromechanical performance of finger sensors at different angles, resistance and time values are recorded using a Bluetooth accelerometer plotter according to the finger movements. The actual angles of the finger were obtained using the Kinovea—0.9.5 program [23].

## 3. Results

### 3.1. Electromechanical Performance of the Knitted Strain Sensor

The electromechanical performance of the knitted strain sensor is investigated and various tensile tests are carried out to investigate the strain sensing properties as illustrated in Figure 3. In Figure 3a,c, the sensor is stretched five times to the range of 40% and the relative resistance change of the sensor is measured for 5 cycles, where R_0_ and ΔR represent the initial and resistance changes, respectively. The first cycle always deviates which was attributed to the first reorganization of the knitted structure and is therefore omitted. The relative resistance changes of the sample show a linear characteristic (R^2^ = 0.98) with the gauge factor (GF) and hysteresis value of 1.19 ± 0.03 and 0.03, respectively. The stability of the sensor is investigated by cycling up to 70 times as shown in Figure 3b. It turns out to be the linearity is maintained but GF drops from 1.19 ± 0.03 to 1.05 ± 0.02 at the 70th cycle.

The static-hold strains during the cycles of the loading and unloading test are measured to demonstrate the retention property of the knitted strain sensor in Figure 4a. In the cycles, the strain was increased with a speed of 30 mm/min to the maximal value and held for 10 s; then it was released to 0% and held again for another 10 s. The fourth cycle of the hold test was selected and illustrated in Figure 4b. During the holding phase, the resistance drops from 0.1633 to 0.1533 (0.01 percentage points) which is attributed to the relaxation of the knitting sensor structure due to the readjustment and sliding of yarns [20].

### 3.2. Application of Knitted Strain Sensor for Rehabilitation Purposes

First, the Version 1 and Version 2 sensor designs were tested on different fingers with the finger bent up to 90 degrees. The pink finger was considered too small for the present sensor versions and was not included in the measurements. The results are summarized in Table 2. The tests were applied to each finger four times and their mean, standard deviations, and coefficients of variation (CV) were listed. The lower the CV value, the more precise and reproducible the results. The index shows the highest resistance change with the lowest variations, and subsequent tests focused on the use of the index finger for sensor performance evaluation.

The sensor performance was evaluated by collecting the electrical resistance and time values at different angles of the finger. For both designs as illustrated in Figure 2, the effect of various angles on resistance change during finger bending and holding was investigated and results are shown in Figure 5. The response of the sensor to the dynamic bending at positions of 45° and 90° is accurate for both demonstrators. A sharp peak in resistance was always observed directly after bending for each version. Where the Version 2 sensor shows resistance changes of 0.05 and 0.04 when the angles change from 0 to 45 and 45 to 90 degrees, the Version 1 design shows resistance changes of 0.17 to 0.18, respectively, as can be seen in Figure 5b,d. These data points were taken for 5 s based on the resistance after the first peaks. The relative resistance changes between the two design versions can be observed to be 3.4 to 4.5 times higher for Version 2 in comparison to Version 1, respectively. For both versions, the peaks are followed by a gradual decrease during holding. This relaxation behavior is commonly observed in textile strain sensors [24]. The signal noise during the holding stage can be explained by small muscle movements and joint twitching [25].

#### The Effect of Different Angles and Spacer Layer Thickness on Finger Movement of Demonstrator Performance

Further tests were applied to Version 1 and the repeated finger movements at 0°, 45°, and 90° were measured for four trials and illustrated in Figure 6. With the increase in the bending angle of the finger joint from 0° to 45° and then to 90°, the change of the relative resistance value gradually increases and it can be tracked with three different signals. The repetitive signals showed that the Version 1 finger sensor can be capable of distinguishing the finger movements while establishing a trend in each finger move. While all tests give distinguishable and similar signals in the 4 trials, after the first bending cycle, the bandwidth between the lower and upper peaks remains reasonably constant for tests 1 and 3. This was not observed in the rest of the tests.

Apart from the repeated tests on Version 1, a further test was also carried out by increasing the spacer layer thickness between the sensor and the finger to observe the effect of the spacer layers on whether the sensor is tightly wrapped with the finger during movement. To obtain a quantitative idea of how much an added spacer layer can increase the strain in the sensor area, we model the finger as two straight beams connected by a cylindrical joint with a diameter of D (see Figure 7). When it bends at an angle alpha (α), the joint surface extends with a length of s = α×D/2 by following the arc length calculation. If a spacer layer with thickness h_*layer*_ is added, the extension of the spacer layer surface simply becomes s = α × (D/ 2 + h_*layer*_). Assuming that the bending is only confined to the joint area and the reference length is defined as an initial length of the part of the sensor, the ε is calculated as in Equation (Equation 1).
(1)$$\varepsilon =\alpha \times (1/2+h_{layer}/D).$$

In the Version 1 sensor design, the thickness of the substrate fabric was 0.1 cm, and the test was carried out by adjusting this thickness up to 0.4 cm by changing the spacer layer number in between. The tests were completed at an angle of α=45 and the results are shown in Table 3.

When the number of spacer layers is increased up to 2, the resistance change tends to increase, while when it is above 4, the resistance change tends to decrease. This is because the ease of finger movement gradually decreases as the number of spacer layers increases, preventing the finger from freely moving. In addition, the resistance change when the number of spacer layers is 2 was found to be 0.40. This is equal to the value obtained at a 90 degree during the normal repetitive test as illustrated in Figure 3b.

## 4. Discussion

Knitted strain sensors integrated into the finger show a change in resistance due to a deformation caused by the finger’s inclination at various angles. This resistance change is due to the elongation of the conductive yarn in the knitted structure, and the contact points between the successive knitted loops of the conductive yarn are separated, causing the sensor to change its resistance [15]. Different sensor designs cause differences in resistance changes. The Version 1 sensor is better than Version 2 in distinguishing different angles at 45 and 90 degrees. This can be explained by the full coverage of the sensor on the finger because of the tubular form. The slackness seen in Version 2 causes the deformation during finger movement not to be fully achieved, which can be a reason for the lower resistance change. When examining the effect of layer thickness, we expected to see an increase in detected sensor signals for each added spacer layer. Each spacer layer with a different thickness added will create a layer between the finger and the sensor. It turned out that as the number of spacer layers increased, the space in between them decreased and it becomes more intact to the finger. However, as the number of spacer layers increases, the ease of finger movement decreases, and this increase in resistance is not observed. The full sensor length is about 50 mm, only the part covering the joint 10 mm is strained, resulting in an overall sensor efficiency of about 20%. The sensor design could therefore be optimized by focusing on the effective areas only.

While the sensor showed a 4.9 percent resistance change after 10 s under a hold-cyclic test as explained in Section 3.1, this situation was also measured when it is integrated into a finger. The resistance change percentages between 0–45 and 45–90 angles are calculated as 16% and 18% for Version 1, 55 and 27% for Version 2 which the difference in resistance is measured after 10 s from the first peak. Version 1 is also a better alternative for keeping the resistance during the holding stage which again can be attributed to the full coverage structure. The fact that the resistance change is measured more than the static test can be explained by the current joint twitching or the effect of small muscles [18].

## 5. Conclusions

We produced finger sensor designs for rehabilitation applications. The previously developed knitted strain sensor [21] is integrated into a finger and the performance of the integrated sensor is tested at different angles of the finger.

The index shows the highest resistance change with the lowest variations for different finger sensor versions. Based on this, all following tests were performed on the index finger.The response of the sensor to the dynamic bending at positions of 45° and 90° is accurate for both demonstrators. However, the resistance change is higher in Version 1 in comparison to Version 2. For both versions, the peaks are followed by a gradual decrease during holding.As the number of spacer layers increases, the change in resistance tends to increase, while as the number of spacer layers increases, the change in resistance tends to decrease. This is because finger movement gradually decreases as the number of spacer layers increases.

## Figures and Tables

**Figure 1 micromachines-14-00710-f001:**
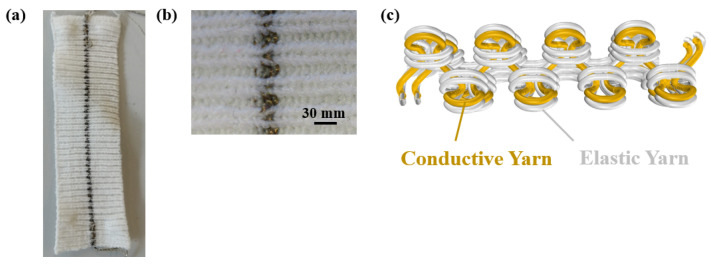
(**a**) The developed knitted strain sensor, (**b**) optical images of the sensing region which shows the conductive yarns (gold) positioned inside and elastic yarns outside (white), and (**c**) illustration of the conductive and elastic yarn positioning within the sensing region.

**Figure 2 micromachines-14-00710-f002:**
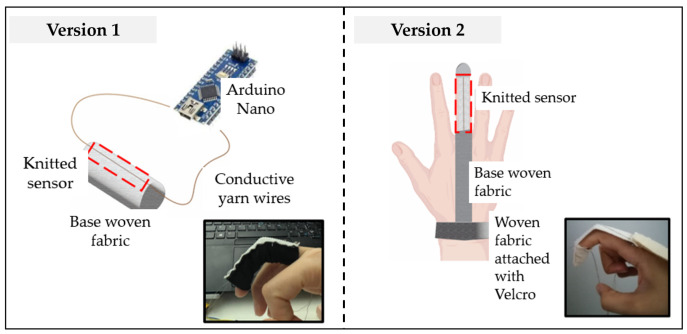
Different versions of wearable finger sensors: (**a**) Version 1 and (**b**) Version 2.

**Figure 3 micromachines-14-00710-f003:**
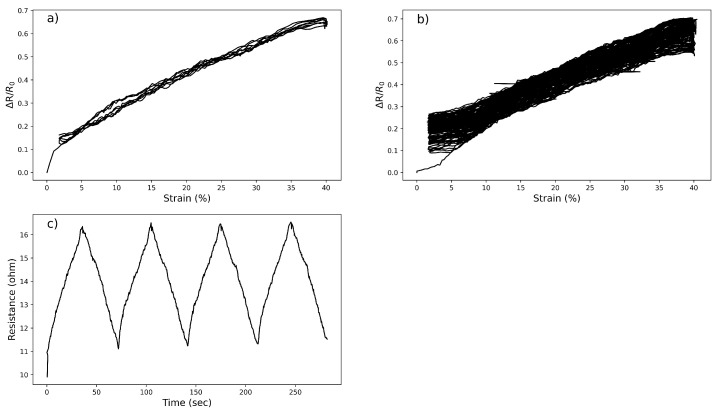
The developed knitted strain sensor graphs under four cyclic tests, (**a**) Relative resistance change versus strain, (**b**) Relative resistance change versus strain for 70 repetitive cycles, and (**c**) Resistance changes versus time at a strain 40% strain under 4 cycles.

**Figure 4 micromachines-14-00710-f004:**
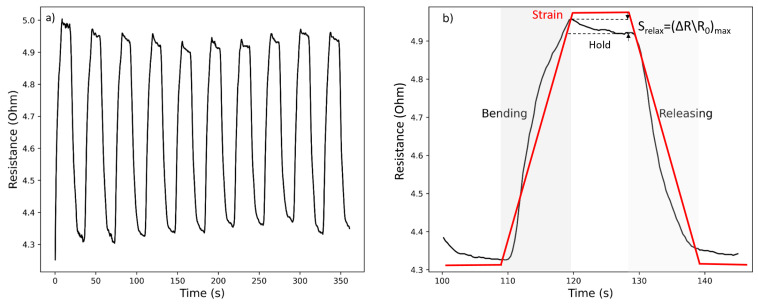
The cyclic hold test of knitted strain sensor: (**a**) Resistance versus time graph when the hold time was set as 10 s, and (**b**) Relaxation during the holding phase.

**Figure 5 micromachines-14-00710-f005:**
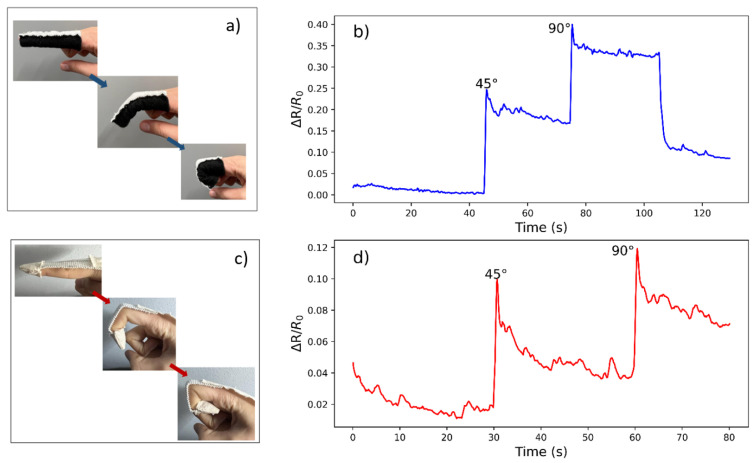
The relative resistance change versus time for Version 1 and Version 2 at various positions: relaxation, 45° and 90°. (**a**) The positions of version 1 sensors at different angles, (**b**) relative resistance changes versus time of version 1 sensor, (**c**) the positions of the Version 2 sensor at different angles, and (**d**) relative resistance changes versus time of the Version 2 sensor.

**Figure 6 micromachines-14-00710-f006:**
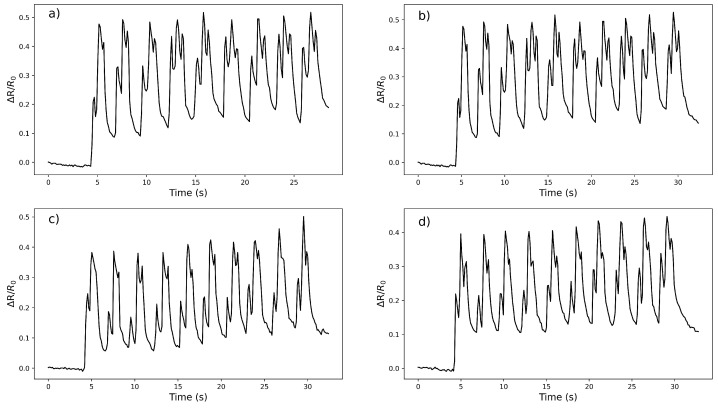
The performance of the Version 1 sensor in four different trials with repeated finger movements 0–45–90 degrees: (**a**) 1.test, (**b**) 2.test, (**c**) 3.test and (**d**) 4.test.

**Figure 7 micromachines-14-00710-f007:**
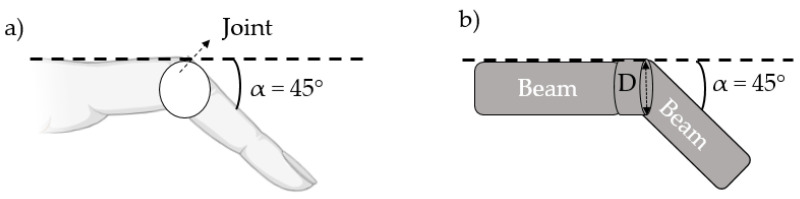
(**a**) The schematic of the index finger during bending at 45 degree and (**b**) representation of the finger as two beams with a cylindrical joint.

**Table 1 micromachines-14-00710-t001:** The advantages and disadvantages of the finger sensor versions.

Design	Advantages	Disadvantages
Version 1	Better coverage to finger	Possibility of slackness after use
Version 2	Easily adjustable to hand size	Discomfort caused by not fully covering the finger

**Table 2 micromachines-14-00710-t002:** Variability of demonstrator relative resistance change measurements for repeatability.

Finger		Version 1			Version 2	
	* **Mean** *	* **SD** *	* **CV (%)** *	* **Mean** *	* **SD** *	* **CV (%)** *
**Thumb**	0.30	0.01	0.05	0.44	0.07	16
**Index**	0.31	0.02	0.06	0.53	0.05	9
**Middle**	0.21	0.04	0.18	0.42	0.06	14
**Ring**	0.25	0.04	0.17	0.42	0.07	17

**Table 3 micromachines-14-00710-t003:** The effect of spacer layer number on resistance changes at an angle of 45° for Version 1 sensor.

Angle	Spacer Layer Number	ε	ΔR/R_0_
45	0	0.39	0.18 ± 0.01
45	1	0.49	0.28 ± 0.02
45	2	0.59	0.41 ± 0.02
45	4	0.80	0.19 ± 0.01

## Data Availability

Not applicable.

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
