# Peer review of "Design of Wearable Finger Sensors for Rehabilitation Applications"

_micromachines, 2023, doi:10.3390/mi14040710_

Round 1

Reviewer 1 Report

Figure 5 it is better to have the units in seconds rather than ms.

A table summarizing the differences from Version 1 and Version 2 sensors is better. 

Author Response

Dear Reviewer,

You can find the point-by-point responses in the attached word file. Note that the sentence highlighted in yellow is integrated into the text.

Regards,

Beyza Bozali

Reviewer 2 Report

The wearable sensor is an emerging technology for conformal integration with human bodies. In this manuscript, Bozali et al. report wearable finger-movement sensors based on a knitted structure. The textile-based sensors can be valuable in health monitoring applications. I recommend the manuscript be considered for publication after addressing the following issues:

1.      I could not quite understand the fabrication process illustrated in Figure 1. Some improvements of the schematic illustration is highly encouraged. A good example is available in Reference [21].

2.      What does the mean value in Table 1 standard for? The values are inconsistent with either Figure 3 (~ 10 ohms) or Figure 5 (ΔR/R0 ~0.4 and 0.12).

3.      The statements about the increasing layer thickness are extremely confusing in the Discussion and Conclusion sections. 

Minor points:

1.      In all figures, the ordinate values should use points instead of commas for decimals.

2.      In Figure 7, the angles are clearly not 45 degrees. 

Author Response

Dear Reviewer, 

You can find the point-by-point answers in the attached word file. Note that the sentence highlighted in yellow is integrated into the text and those in red are the responses to the points.

Regards,

Beyza Bozali
